EMBO
Molecular Medicine

# Fair priority setting in gene-editing research: the accountability for reasonableness framework

Marije J Smits[1], Manon van Daal [2✉], Marcel Verweij[3], Edward Nieuwenhuis[4], Sabine Fuchs[5,6] & Karin R Jongsma[2,6]

**With rapid scientific advancements in gene-editing technologies, we advocate using the Accountability for Reasonableness (A4R) framework to guide priority setting of gene-editing research, ensuring equitable, efficient, consistent, and accountable translation of these revolutionizing innovations into society.**

As precise gene-editing and delivery technologies in humans advance rapidly, their potential to treat a wide range of previously intractable genetic disorders becomes increasingly tangible. This expanding therapeutic landscape brings challenges to the forefront, particularly regarding how to prioritize diseases for research and subsequent clinical application. Currently, prioritization is largely guided by technical feasibility, with little attention to broader ethical conditions and transparent selection criteria. A recent real-world example highlights the urgency of addressing these concerns. On May 15, 2025, Musunuru reported the first personalized gene correction therapy (Musunuru et al, 2025) in a patient with Carbamoyl Phosphate Synthetase I (CPS1) deficiency, a rare inherited metabolic disorder. This case marks a major milestone in using personalized gene editing as a transformative healthcare technology to precisely correct pathogenic DNA variants. The patient's selection, based on technical feasibility, such as a base-editable mutation without clinically relevant bystander editing and the

liver-specific nature of CPS1, was appropriate for early-stage development. However, as gene-editing technologies continue to expand in possible applications, it becomes increasingly important to determine how to prioritize genetic diseases for gene-editing research in an ethically sound manner.

An ethical framework for setting priorities in gene therapy research is currently missing but is highly needed, as it is essential to determine which patient groups will, and which will not, have access to these innovations. This is fundamentally a matter of health justice. Furthermore, gene therapies are highly expensive and as more applications become available these will negatively impact the financial sustainability of public health care systems (Vallano and Pontes, 2024). Health systems are increasingly under pressure meaning that even novel gene therapies that have proven effective may not be reimbursed by health insurances. Fair priority setting will be crucial for the future of gene-editing research.

Currently, decisions about research priorities at most labs and research organizations are based on technical feasibility, the specific expertise of research groups and funding possibilities. Since prioritization choices in gene-editing research shape availability of gene-editing therapies, these decisions should be based on a fair and consistent procedure to maximize impact. A transparent, consistent and fair process is

more trustworthy, which can contribute to public trust (Hudson et al, 2016), prevent discrimination and protect minority rights by providing equal treatment under codified standards.

We believe there is an urgent need to implement fair and transparent decision-making frameworks for gene-editing priority setting as the technology is approaching its full potential. The *Accountability for Reasonableness* (A4R) framework, originally developed for healthcare priority setting, provides a suitable model to guide equitable selection of diseases (and hence patients) in gene-editing research.

## Accountability for reasonableness applied to gene editing

A4R, developed by Norman Daniels and James Sabin (Daniels, 2001), consists of four key conditions for promoting accountability and legitimacy of healthcare priority setting.

*Publicity*, the first condition, requires that decisions in priority setting, along with their rationales, are publicly accessible (Daniels, 2001). This kind of transparency ensures that everyone affected by the decision has access to and can understand the reasoning behind the decision. Applied to research priority setting it means that: *decisions about which research areas/diseases receive funding or attention should be made transparent.* Although trial registries list ongoing gene-editing studies, data on disease selection, funding criteria, and

[1]Department of Pediatrics, Division of Metabolic Disorders, Emma Children's Hospital, Amsterdam UMC, University of Amsterdam, Amsterdam, Netherlands. [2]Department of Bioethics and Health Humanities, Julius Center for Health Sciences and Primary Care, University Medical Center Utrecht, Utrecht University, Utrecht, The Netherlands. [3]Ethics Institute, Utrecht University, Janskerkhof 13a, Utrecht 3512 BL, The Netherlands. [4]Princess Máxima Center for Pediatric Oncology, Utrecht, The Netherlands. [5]Division of Metabolic Diseases, Wilhelmina Children's Hospital, University Medical Center Utrecht, Utrecht, the Netherlands. [6]These authors contributed equally: Sabine Fuchs, Karin R Jongsma. ✉E-mail: m.vandaal@umcutrecht.nl
https://doi.org/10.1038/s44321-026-00420-w | Published online: 8 April 2026

rationales are often incomplete. In addition, transparency should also cover the uncertainties behind decisions made.

Consider, for example, the following hypothetical case: a research group working in an academic hospital faces a decision between three pediatric patients, each with a rare liver disorder caused by a single gene mutation, and all equally ideal candidates for developing personalized base-editing therapy. Technically, each case is a "perfect" candidate, yet limited resources allow treatment for only one. The research group wants to justify their decision and turns to the A4R framework for guidance.

Applying the *publicity condition* means that both researchers and funders should disclose the reasons for prioritizing one disease over another. Researchers should also assess whether funding agencies make their research agendas or criteria, such as disease burden, equity, or under-researched populations, publicly available. Alternatively, researchers can explicitly motivate their rationale for prioritization and the composition of the deliberating committee. Making such information public helps build a 'case law' of reasoned decisions, enhancing accountability and informing future research directions (5).

The *relevance condition* entails that priority setting decisions should be built on a "reasonable construal" of principles (Daniels, 2001). In this context, 'reasonable' means that fair-minded people, including patients, researchers, policymakers, and industry representatives, can agree that the principles and evidence are relevant for addressing health needs fairly within resource limitations. Fair-minded refers to those that "seek mutually justifiable terms of cooperation" (Daniels, 2001). If fair-minded stakeholders are involved in identifying relevant considerations, those disadvantaged by a priority-setting decision can be assured it was based on fair and pertinent criteria. Applied to research priority setting it means that *the reasons for selecting research priorities must be seen as reasonable by all relevant stakeholders*.

Applying the relevance condition to our fictional case, the condition requires the research group to deliberate on decisions about what to study, who benefits, and how resources are allocated. These decisions should be grounded in reasons that are not only scientifically or strategically justifiable, but also ethically and socially. Priority setting may include burden of disease,

neglect or inequity (e.g., underfunded conditions), scientific feasibility, social impact, and global or local health priorities. In this endeavor, it is important that the research group deliberates with relevant stakeholders, including those affected by the decision, to determine which of these criteria are most appropriate.

A critique of the relevance condition is the limited possibility to address inclusivity and potential power imbalances among stakeholders (Manafò et al, 2018). Since dominant actors may disproportionately influence decisions, some suggested to add an explicit *empowerment condition* to A4R, emphasizing inclusive participation and incorporation of community values in decision-making (Gibson et al, 2005).

The *revisability condition*, also known as 'appeals criterion', refers to the required mechanisms to challenge and adjust decisions when new evidence or arguments emerge (Daniels, 2001). This allows for learning by experience and correction of previous potential errors. In the context of research priority setting, it means that *stakeholders should have avenues to contest or question funding decisions or research agendas, especially as societal needs, values, or scientific knowledge evolve.*

The revisability condition seems particularly relevant for gene-editing research, where rapid technological and clinical developments can quickly change the landscape. New evidence, such as changes in technical feasibility, disease epidemiology, burden of disease, availability of alternative therapies or reduced development costs may significantly influence research priority setting considerations. To apply the revisability condition to our case, researchers should remain open to revising their choices and critically reassess priorities when credible concerns arise. Senior researchers in leadership or advisory roles should in addition help design institutional processes that enable meaningful revision. While revising funding decisions midstream can be difficult, the decision-making process itself could be reviewed by an external party to assess adherence to procedural justice. This evaluation can then be used to revise the A4R framework for a next funding round issued under these conditions. This approach promotes ongoing learning and helps ensure that research priorities remain responsive to advances in gene-editing technologies.

Finally, the *enforcement condition* entails that voluntary or public regulations are in place to ensure that the above-mentioned conditions are met (Daniels, 2001). In the context of health research priority setting, this condition is essential for *safeguarding procedural fairness and accountability in how research agendas are shaped, particularly when public resources, marginalized populations, or ethically sensitive technologies (such as gene editing) are involved.* Unlike health care priority setting, where decisions are often directly enforced through institutional policies or governmental regulation, enforcement in research contexts is more fragmented among funders, institutions, researchers, pharmaceutical companies and ethics boards.

While researchers are not typically enforcement agents, they may play a crucial enabling role in ensuring enforcement of the A4R framework by applying the described conditions in their own practices, advocating for their institutions to adopt these principles and complying with oversight. To apply the enforcement condition to our case, the research group may anticipate their responsibility by outlining how they comply with ethical oversight, describe how affected communities are included and how they will share good practices with the wider research community. Meeting the enforcement condition could also involve external reviews, audits of prioritization processes, inclusion of procedural criteria in funding assessments or evaluation by (international) policy making by healthcare institutions.

## Constraints of the A4R approach in gene-editing research

While A4R offers a structured approach to procedural justice, it faces several challenges. First, the focus on fair processes does not guarantee fair outcomes; transparency and inclusiveness may thus still lead to unjust results (Nunes and Rego, 2014). Second, the effectiveness of A4R depends on fair-minded participants capable of identifying reasonable criteria. In emotionally charged contexts, such as rare, untreatable diseases, urgency may compromise impartial deliberation. For example, emotional appeals from parents have influenced the prioritization of ultra-rare disease research (Ledford, 2024), raising concerns about the integrity of A4R guided processes under such pressures. Third, because most

gene-editing research occurs in the Global North and depends on proximity to patients, A4R does little to counter regional biases, potentially favoring diseases affecting more vocal or better-resourced populations. Fourth, fully adhering to all four conditions of the A4R framework may appear impractical due to the time required to engage diverse stakeholders and integrate their perspectives. Finally, A4R is a framework that ensures that those affected by gene-editing technologies are included in the decision-making process *of* research groups and institutions. Note, however, that priority setting is also necessary in the wider allocation of public funding for scientific research, and such decisions should ultimately be governed by policies that are shaped in democratic procedures. A4R might also enhance democratic legitimacy of those funding decisions, but arguably more extensive forms of public deliberation and citizen engagement would be appropriate.

While all these concerns are valid, we argue that implementing A4R to promote explicit justification and structured priority setting may, over time, lead to greater efficiency and time savings in research institutions. A key condition for implementation of the A4R framework is (international) recognition of its necessity and commitment to installing associated procedures.

## Conclusion

Sustained rapid progress in gene-editing technology demands agility of researchers, ethicists and policy makers to allow for responsible allocation of gene-editing research and therapy development efforts. In the current absence of prioritization methods, A4R presents a guide for responsible priority setting in gene-editing research. A4R holds deliberation and transparency essential in the decision-making process. Adhering to the four conditions of A4R enables researchers to incorporate diverse values. While democratic processes may determine overarching values and goals in public funding of scientific research, A4R can guide fair decision-making within those democratically established parameters. This framework empowers researchers and clinicians to take responsibility for fairness in their own institutional contexts and allows them to actively shape and contribute to ethical deliberation on priority setting in gene-editing research. A4R enhances the legitimacy and trustworthiness of decisions in research organizations, and thereby acceptability for those affected by the decisions. This may serve as a foundation for more equitable and accountable practices in the rapidly advancing field of gene-editing research. Further research should focus on the feasibility of A4R for gene-editing priority setting in research, as well as the implications of each of the four conditions.

## Peer review information

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

## Acknowledgements

This work is part of KNAW Ammodo Science Award for Groundbreaking Research 2022 in the domain of Biomedical Sciences ("From cutting-edge organoid and gene editing technologies to in vivo gene correction in patients with a genetic disease").

## Disclosure and competing interests statement

The authors declare no competing interests.

