## [Peer Review File · EMBO Molecular Medicine]

Fair priority setting in gene-editing research: the Accountability for Reasonableness Framework

Marije .J. Smits, Manon van Daal, Marcel Verweij, Edward Nieuwenhuis, Sabine Fuchs, Karin .R. Jongsma

Corresponding author: Manon van Daal, (m.vandaal@umcutrecht.nl)

Review Timeline:

Submission Date:	25th Nov 25
Editorial Decision:	14th Jan 26
Revision Received:	18th Mar 26
Accepted:	18th Mar 26

Editor: Zeljko Durdevic

Transaction Report:

14th Jan 2026

Dear Dr. van Daal,

Thank you for the submission of your manuscript to EMBO Molecular Medicine. I am pleased to inform you that we will be able to accept your manuscript pending the following final amendments:

- 1) Please implement the referee's suggestion and include discussion about other possible means of just priority-setting.
- 2) Please check our Author Guidelines for more information about the manuscript formatting.
<https://link.springer.com/journal/44321/submission-guidelines#cms-Article-types>
- 3) Add "Disclosure Statement & Competing Interests". We updated our journal's competing interests policy in January 2022 and request authors to consider both actual and perceived competing interests. Please review the policy
<https://www.embopress.org/competing-interests> and update your competing interests if necessary.
- 4) Author contributions: Please specify author contributions in our submission system. CRediT has replaced the traditional author contributions section because it offers a systematic machine-readable author contributions format that allows for more effective research assessment. You are encouraged to use the free text boxes beneath each contributing author's name to add specific details on the author's contribution. More information is available in our guide to authors:
<https://www.embopress.org/page/journal/17574684/authorguide#authorshipguidelines>
- 5) Rename "List of literature" to "References" and correct the reference citation in the text and reference list. In the text a reference should be cited by author and year of publication. Include a space between a word and the opening parenthesis of the reference that follows. In the reference list, citations should be listed in alphabetical order. Where there are more than 10 authors on a paper, 10 will be listed, followed by "et al.". Please check "Author Guidelines" for more information.
<https://www.embopress.org/page/journal/17574684/authorguide#referencesformat>
- 6) As part of the EMBO Publications transparent editorial process EMBO Molecular Medicine will publish online a Review Process File (RPF) to accompany accepted manuscripts. This file will be published in conjunction with your paper and will include the anonymous referee reports, your point-by-point response and all pertinent correspondence relating to the manuscript.

You can submit your revised files by logging onto our online manuscript tracking system or simply follow this

I look forward to receiving the revised version of your manuscript as soon as possible.

Yours sincerely,

Zeljko Durdevic

Zeljko Durdevic
Senior Editor
EMBO Molecular Medicine

*** IMPORTANT INFORMATION ***

- 1) a .doc formatted version of the manuscript text (including Figure legends and tables)
- 2) Separate figure files
- 3) a letter INCLUDING the reviewer's reports and your detailed responses to their comments.

Also, and to save some time should your paper be accepted, please read below for additional information regarding some features of our research articles:

1) Disclosure and competing interest statement: Please include a statement declaring any competing commercial interests in relation to your submitted work.

2) Please note that we now mandate that all corresponding authors list an ORCID digital identifier. This takes <90 seconds to complete. We encourage all authors to supply an ORCID identifier, which will be linked to their name for unambiguous name identification.

Currently, our records indicate that the ORCID for your account is 0000-0003-1718-2827.

Link Not Available

-

Thank you,

Zeljko Durdevic

***** Reviewer's comments *****

Referee #1 (Remarks for Author):

"Fair priority setting in gene-editing research: the Accountability for Reasonableness Framework" is a fairly polished commentary arguing for the implementation of the A4R framework, which was originally developed for healthcare priority setting, to the context of gene-editing research, for the purposes of fairly prioritizing targets for research (i.e., conditions). While I commend the author/s in their adept explanation and application of the conditions of publicity, relevance, revisability, and enforcement to gene-editing research, they fail to discuss other possible means of just priority-setting. For instance, what about deliberative public engagement, which invokes the importance of democratic principles? The commentary seems focused on a more micro-level of priority setting, perhaps at an institutional level, but it would seem that such a framework would be entirely pluralistic, depending on which stakeholders are the decision makers. How would such a micro-level framework engage with more macro levers of governance?

Editor Comments to Author:

1. "Fair priority setting in gene-editing research: the Accountability for Reasonableness Framework" is a fairly polished commentary arguing for the implementation of the A4R framework, which was originally developed for healthcare priority setting, to the context of gene-editing research, for the purposes of fairly prioritizing targets for research (i.e., conditions). While I commend the author/s in their adept explanation and application of the conditions of publicity, relevance, revisability, and enforcement to gene-editing research, they fail to discuss other possible means of just priority-setting. For instance, what about deliberative public engagement, which invokes the importance of democratic principles? The commentary seems focused on a more micro-level of priority setting, perhaps at an institutional level, but it would seem that such a framework would be entirely pluralistic, depending on which stakeholders are the decision makers. How would such a micro-level framework engage with more macro levers of governance?

Author Response:

We thank the reviewer for this comment. A key point raised by the reviewer concerns alternatives for the A4R framework, including deliberative public engagement grounded in democratic principles. The primary aim of our paper was to provide clinicians and researchers with a practical framework to promote and enhance fairness in gene-editing research. We focus on decision-making within laboratories and research organizations. In response, we have clarified in the revised manuscript on page 3 that while A4R is proposed as a framework for fair decision-making within research groups and institutions, priority setting in the broader allocation of public research funding should ultimately be governed by democratically shaped policies and regulations. We now explicitly acknowledge that, at that level, more extensive forms of public deliberation and citizen engagement may be appropriate. This addition clarifies that A4R is not intended as a substitute for democratic processes, but as a procedural framework suited to institutional contexts, which can operate within democratically established parameters.

The reviewer is also correct that the commentary primarily addresses priority setting at a micro or meso level. We chose this focus to address an area where decisions could benefit from fair procedures as this gap has been explicitly highlighted by stakeholders in our center, such as researchers and clinicians working on gene-editing technologies. At the macro level, democratic processes may determine overarching goals, funding envelopes, or regulatory constraints. Within those democratically established parameters, A4R can guide fair decision-making. In this way, A4R serves as a bridge between micro-level priority setting and macro-level governance. We addressed this in the conclusion on page 4.

18th Mar 2026

Dear Dr. van Daal,

We are pleased to inform you that your manuscript is accepted for publication and is now being sent to our publisher to be included in the next available issue of EMBO Molecular Medicine.

Your manuscript will be processed for publication by EMBO Press. It will be copy edited and you will receive page proofs prior to publication. Please note that you will be contacted by Springer Nature Author Services to complete licensing information.

Zeljko Durdevic
Senior Editor
EMBO Molecular Medicine